# Telehealth and COVID-19 Pandemic: An Overview of the Telehealth Use, Advantages, Challenges, and Opportunities during COVID-19 Pandemic

**DOI:** 10.3390/healthcare10112293

**Published:** 2022-11-16

**Authors:** Khayreddine Bouabida, Bertrand Lebouché, Marie-Pascale Pomey

**Affiliations:** 1Centre de Recherche du Centre Hospitalier de l’Université de Montréal (CRCHUM), Montreal, QC H2X 0A9, Canada; 2École de Santé Publique, Département de Gestion, D’évaluation et de Politique de Santé, Université de Montréal, Montreal, QC H3N 1X9, Canada; 3Department of Family Medicine, Faculty of Medicine & Health Sciences, McGill University, Montréal, QC H3A 0G4, Canada; 4Centre for Outcomes Research & Evaluation, Research Institute of the McGill University Health Centre, Montréal, QC H4A 3J1, Canada; 5Chronic Viral Illness Service, Department of Medicine, Division of Infectious Diseases, McGill University Health Centre, Montréal, QC H4A 3J1, Canada; 6Infectious Diseases and Immunity in Global Health Program, Research Institute of the McGill University Health Centre, Montréal, QC H4A 3J1, Canada; 7Hospital Center of the University of Montreal (CHUM), Montreal, QC H2X 0C1, Canada; 8Centre of Excellence on Partnership with Patients and the Public, Montreal, QC H2X 0A9, Canada

**Keywords:** telehealth, digital health, health technology, COVID-19, pandemic, impacts, care delivery, care access

## Abstract

The use of telehealth and digital health platforms has increased during the COVID-19 pandemic due to the implementation of physical distancing measures and restrictions. To address the pandemic threat, telehealth was promptly and extensively developed, implemented, and used to maintain continuity of care offered through multi-purpose technology platforms considered as virtual healthcare facilities. The aim of this paper is to define telehealth and discuss some aspects of its utilization, role, and impact, but also opportunities and future implications particularly during the COVID-19 pandemic. In order to support our reflection and consolidate our viewpoints, numerous bibliographical sources and relevant literature were identified through an electronic keyword search of four databases (PubMed, Web of Science, Google Scholar, and ResearchGate). In this paper, we consider that telehealth to be a very interesting approach which can be effective and affordable for health systems aiming to facilitate access to care, maintain quality and safety of care, and engage patients and health professionals and users of health services. However, we also believe that telehealth faces many challenges, such as the issue of lack of human contact in care, confidentiality, and data security, also accessibility and training in the use of platforms for telehealth. Despite the many challenges it faces, we believe telehealth has enormous potential for strengthening and improving healthcare services. In this paper, we also call for and encourage further studies to build a solid and broad understanding of telehealth challenges with its short-term and long-term clinical, organizational, socio-economic, and ethical impacts.

## 1. Context

The COVID-19 pandemic caused many tragic effects worldwide and seriously affected the regular performing capacity of the healthcare systems. Due to the absence of proper vaccines and treatments in the early stages of the pandemic, approaches such as lockdown, quarantine, and physical distancing were implemented to slow down the spread and reduce the impact of the disease [1,2]. However, these measures induced multiple negative socio-economic effects but also effects on populations’ health such as difficulty accessing the care systems, depression and isolation anxiety, etc., affecting the quality of care and life [3,4]. Thus, the COVID-19 pandemic acted as a catalyst for the development and adoption by health systems, especially of industrialized countries, of a broad range of telehealth technologies, developing and innovating in new virtual health facilities and telecare platforms to minimize these effects [3,4,5,6,7]. In this context, we opted to formulate a reflexive paper in which we present an overview to first define telehealth and discuss its utilization, role, and impact, particularly during the COVID-19 pandemic. In this paper, we also discuss certain telehealth perspectives, opportunities, and future implications. Note, that as we formulated our reflection on this topic and to support our viewpoint, we also consulted numerous bibliographical and literature sources using appropriate keywords related to the topic for search engines such as Google Scholar, PubMed, ResearchGate, etc. Broad search terms were used for their relevance to the paper’s viewpoint such as Telehealth OR telecare OR telemedicine OR e-health OR mobile health apps AND the COVID-19 pandemic. However, the search was confined to peer-reviewed papers, as our goal in this paper was not a literature review or a systematic review, so unpublished papers and grey literature were not considered appropriate to use for supporting our viewpoint in this paper.

### 1.1. Telehealth Definition and Concepts

As a definition, the term telehealth is associated with a multitude of expressions such as “e-health”, “m-health”, cyber health, or digital health. It is defined as “the combined use of the internet and information technology for clinical and organizational purposes, both locally and remotely” [7,8,9,10]. The field of telehealth is very broad and includes telemedicine, telecare, tele-expertise, teleconsultation, and telemonitoring/remote monitoring but also other tools, such as robotics and artificial intelligence and mobile health (m-health), which include connected objects and smartphone applications [6,7,8,9,10]. Since 2005, the WHO has considered m-health as a priority in the fight against health inequalities and access to care, particularly in developing nations. As a concept, telehealth is new and comprised of virtual healthcare facilities that allow communications and exchange of health information using mobile smart connected devices and information technology to provide virtual and remote care instead of traditional care [7,8,9,10]. Classifications related to telehealth may vary depending on the country and the standards of the health systems. However, often a three-level classification is used based on the functionality and features associated with the technologies: (1) the technologies that allow services to be offered without any patient self-measurement data; (2) the technologies that allow an exchange of information, communication, or performance of simple measurements and data related to well-being; and (3) the remote monitoring technologies aimed at monitoring and processing the clinical data of patients related to their state of health or their disease for the purpose of monitoring and guidance to help clinical decisions and medical diagnosis [6,7,8,9,10,11,12]. Even if telehealth has experienced a tremendous rise during the last decade but especially in the last two years due to the pandemic context, some telehealth models and technologies for delivering healthcare remotely existed prior to the COVID-19 pandemic. The idea of telehealth dates back more than a century [11,12]. Telehealth has been in existence since the 1870s when letters to the editor of the Lancet emphasized the benefits of telehealth [10,11,12]. In a letter published in 1878, ABM advocated using the telephone to illustrate and examine the sound produced by muscles during contraction. They then wondered if the telephone would be superior to the stethoscope [11]. Furthermore, telehealth’s application to deliver medical expertise to areas lacking access to treatment in rural and underserved areas began in the 1960s at Massachusetts General Hospital, delivering care to 1000 patients [12]. In March 2020, within a few weeks, telehealth was at the forefront of and was a critical method for offering care services.

### 1.2. Advantages and Benefits

The use of telehealth has been of global interest among scientific communities even before the COVID-19 pandemic, as demonstrated by several studies [11,12,13,14,15,16]. There are many expectations related to telehealth, and patients could greatly benefit from their use. Studies show that the use of connected technologies makes it possible, for certain patients suffering from chronic diseases (asthma, diabetes, hypertension, etc.), to improve their state of health and their quality of life, reduce their symptoms, and strengthen their self-management skills [13,14]. These technologies would thus contribute to improving patient engagement, and their adherence to good lifestyle habits, to their treatment and to equipping them to maintain their autonomy. Studies have shown that telehealth offers patients the possibility of collecting and monitoring, in real-time, a set of data specific to their health or their diseases without the physical presence and direct intervention of health professionals [15,16]. Furthermore, telehealth platforms also allow patients to communicate and interact virtually with their health professionals and to obtain advice or information relevant to their health [14,15,16,17,18,19]. A patient who is well-informed about their disease, their state of health, and their treatments turns out to be better engaged in the management of their health and would benefit from a better quality of life [15,16]. Telehealth could therefore play an important role in this process, alongside other things, thanks to the data and information that telehealth collects and exchanges between patients and health professionals. This exchange of information would generate new shared knowledge and collective wisdom between patients and professionals, which would strengthen patients in their self-management capacities and engagement in their care trajectory [15,16,17,18,19]. Telehealth could also, in the context of good use and subject to its reliability, facilitate and strengthen the patient/professional relationship by allowing doctors and patients to make better informed and more enlightened decisions. For healthcare professionals, easier access to relevant information enables them to be better equipped during medical consultations and patient follow-up, supports them in decision-making and improves the exchanges and communication with patients [14,15,16,17,18,19].

In light of the potential of telecare facilities, the extensive application of telehealth facilities is not far from expectations. Numerous scientific papers and publications show the significance of the use of telehealth worldwide [20]. Furthermore, in health systems [20], telehealth could improve access to healthcare facilities over distance and outside of working time [19,20,21] and can be an alternative solution to lower hospital admissions for acute conditions [20]. Telehealth services might substantially improve patients’ care and treatment, specifically in areas where there are limited resources [20], and structures, or with a dense population where providing adequate care is challenging, and access to healthcare facilities is limited. As a demonstration, in China, at the beginning of the pandemic, a higher fatality rate of COVID-19 was shown in those areas with limited resources and access to health compared to the areas with adequate access to healthcare facilities [20,21]. Then, when telehealth and COVID-19 remote monitoring platforms were introduced, these areas quickly reported a decline in the mortality rate [20,21]. Studies have been carried out to evaluate the benefits of the use of telehealth during the COVID-19 pandemic in numerous medical fields such as dermatology, cancer, psychology, etc., providing evidence that telehealth use within those specialties helped effectively reduce the transmission of COVID-19 [22,23].

Moreover, the use of technology in telehealth has several benefits, mainly in non-emergency situations or routine care, where physical examinations and interactions between patients and care providers are not required, such as in psychological or mental health services [23,24]. Telecare reduces the resources used in healthcare facilities, minimizes infection transmission, and improves access to care [24,25]. In fact, in the emergency state, as it was during the beginning of the COVID-19 pandemic, telehealth heath was mainly used as a safety barrier for keeping patients, healthcare professionals, and the general public protected while continuing to provide access to care remotely [25,26]. Hence, telehealth technology acts as an effective, attractive, and affordable option, and people are often willing to use it [26,27,28].

Telehealth demonstrated improvements in the efficacy of therapeutic interventions and improvements in the quality of care while also providing patients with psychological support, helping them save time, enhancing treatment compliance, and saving money [29,30,31]. Telehealth lowered costs and improved patient and provider convenience by reducing travel time. Miller et al. [30] corroborated these benefits in their study on the adoption of telemedicine physical therapy services in the context of the COVID-19 pandemic. The authors conducted 4548 physical therapy sessions at the beginning of the COVID-19 pandemic remotely, using a telehealth platform over three months. Then, they conducted a survey to evaluate patients’ perceptions and found that 94% of participants were satisfied with the outcome of the sessions, and 92% would attend a physiotherapy session remotely even in a non-pandemic context [30,31]. Telehealth use also showed that it can improve the learning and communication skills of healthcare professionals. It has been suggested that using telehealth during the COVID-19 pandemic improved the acquisition of better communication competencies for health professionals and after using telehealth platforms for a long period, health professionals now feel more confident communicating even in challenging organizational contexts especially when hospitals are short on supplies resources [30,31,32,33]. Finally, by improving communication in particular “Intrahospital communication”, telehealth can eventually improve and reinforce patient safety [31,32,33,34,35]. From our perspective, we believe that when doctors, staff, and patients are not effectively sharing information, the efficiency of each process may decrease, potentially resulting in unnecessary costs or even danger to patients. Patient record delays, lack of procedural coordination, and even serious medical errors may all be consequences of poor intrahospital communication which might be supported and reinforced thanks to the adequate use of telehealth [31,32,33,34,35].

### 1.3. Limits and Challenges

As in any new emerging innovative domain, limitations exist. The question of maintaining human contact and the pertinence of the virtual vs. in-person care process has been often raised by patients, healthcare professionals, and researchers. One may believe that patients prefer to see their primary care provider in person rather than online or virtually and sometimes with someone with whom they have had no prior contact. In addition, at a time of need, patients tend to return to what they were classically used to i.e., interacting with healthcare professionals in person [35,36]. The quality of care in terms of patient engagement, empathy, and emotional and human consideration can be affected during virtual patient interactions with healthcare professionals on a telehealth platform [36,37]. Others are worried about the practical and clinical safety, quality, confidentiality, and data security [37,38,39]. Indeed, some may consider telehealth to be a threat due to a prospective weakening of therapeutic relationships, reduced stability of care, and compromised confidentiality [38,39,40]. Additionally, the use of telecare in rural areas and among patients with a lower income is inadequate. Lack of resources and supplies, the accessibility of medical facilities, and internet access result in the variation in the utilization of telehealth [40,41,42,43]. Other important limits are that the income and social-economic status of telehealth users affect the effectiveness and adequate use of telehealth platforms [43,44]. Patients of a low socio-economic status may not fully benefit from the use of telehealth care services as those with higher social-economic status do and their experience might be even worse than the classical use of health services. This can be explained by education level and understanding of the basics of connected devices’ technologies, features, and functionalities of telehealth platforms but also the absence of financial means to provide access to the telehealth devices and an adequate internet network [43,44]. Finally, certain experts raise various malpractice liability concerns including informed consent, rules and procedures for treatments that meet the classical and appropriate standard of care, supervision of care providers, and the availability of professional liability insurance coverage specific to telecare use [43,44,45]. As telehealth usage develops, so will the risk of fraud and misuse, necessitating rigorous legislation to keep the operations legitimate and correct [44,45].

### 1.4. COVID-19 Impacts on Telehealth

As a first and major impact, the COVID-19 pandemic accelerated the implementation and use of telehealth at a worldwide level [46]. In fact, before the COVID-19 pandemic, only a few telehealth services were allowed in medical practice, and they were constrained by strict regulations and very few of them were eligible for health insurance and reimbursement [47]. All of this changed after March 2020, when waivers were given for telehealth requirements, and all beneficiaries were given the right to use telecare services anywhere, even in their own homes. Telecare was already in use in the United States prior to the COVID-19 era; however, the pandemic resulted in a massive surge from 13,000/week to 1.7 million/week post-COVID-19 [47,48]. Koonin et al. described an over 13,000% growth in telehealth visits in practically all sectors and specialties in October 2020 [48,49,50]. In Australia, a national study suggests that the COVID-19 pandemic considerably encouraged and promoted the use of telehealth and thanks to this window of opportunity, a lot of progress was made in the domain that, without that progress, telehealth use would have remained infrequent [50,51,52,53,54,55,56,57,58]. In China, telehealth use increased dramatically during the beginning of the pandemic in the country’s major hospitals [50]. The massive use of teleconsultations and tele-triage for suspected COVID-19 cases for mild symptomatic patients relieved a significant burden on healthcare facilities [50]. Again in China, a study among physicians showed a 94.6% use of telemedicine during the pandemic, with 34.1% of them having never used telemedicine before; additionally, they believed that they would never have thought of using telehealth without the obligations of COVID-19 and they were willing to continue using it [50,51]. During the COVID-19 pandemic, Canada also saw a rise in the use of telehealth services. According to a study, the number of people using telehealth in Ontario went from 1.6% in the second quarter of 2019 to 70.6% in the second quarter of 2020 [51]. In another study, conducted in British Columbia, the number of teleconsultations with internal medicine physicians went up from 10% to 80.7% [49]. Another pan-Canadian study illustrated a significant increase in the use of telehealth technologies where phone consultations grew from 43.9% prior to the COVID-19 to 97.6% during the pandemic; meanwhile, virtual online consultations increased from 19.3% prior to the pandemic to 41.2% during the pandemic [52].

Before the COVID-19 crisis, in the USA, private coverage for telehealth insurance was different depending on different plans and was different from payer to payer. In addition, according to a survey, 75% of people with behavioral health problems in the USA continued to receive therapy because of the accessibility of telehealth facilities. However, a delay in regular healthcare was reported by 42% of people [53]. Globally, studies have shown that during the COVID-19 pandemic, females used telecare facilities two times more than males and suggested that young adult patients were represented as the most frequent telecare service consumers, from 20 to 44 years of age [53,54,55,56,57]. In general, studies on the satisfactory level of use of telehealth services during the COVID-19 pandemic suggested it was above average by patients and health professional users and often they showed their willingness to continue with the use of telecare even after the pandemic [56,57,58,59,60,61,62]. Finally, it is difficult to estimate the number of patients spared from contamination cases thanks to the use of telehealth and remote monitoring platforms; however, studies in the United States, Switzerland, and Germany suggest that telehealth has tremendously helped reduce and slow down the spread of coronavirus [62,63,64,65,66]. Thanks to the rapid and massive introduction and use of telehealth platforms, the number of cases and deaths would have been worse globally without telehealth use [66,67,68,69].

During COVID-19, telehealth has played a direct role in lowering the infection rate by tracking symptoms, facilitating physical distancing, encouraging policymakers to foresee the needs of people, and choosing proper strategies and suitable health interventions [70]. The COVID-19 pandemic boosted and pushed healthcare systems and people to evaluate what is feasible and necessary together with adapting standards of care during the rapidly growing pandemic situation [71]. Now, even with the decline in COVID-19, several countries are still investing more and more in the development and promotion of telehealth to prevent the chaos we saw in the early beginning of the pandemic in 2020 and to better manage any further global health crises [71,72,73].

### 1.5. Trends, Opportunities, and Future Implications

We believe that there are three main trends that are currently shaping telehealth. The first is the transformation of the application of telehealth from increasing access to healthcare to providing flexible and convenient care services and eventually reducing the cost of the care services [73,74,75,76,77]. The second is a transformation or “shifting” of telehealth use from hospitals and clinical settings to the home and mobile connected-devices for remote monitoring [73,74,75,76,77]. The third, which in our perspective is the most interesting trend, is the expansion of telehealth from addressing acute conditions to also addressing episodic and chronic conditions [73,74,75,76,77].

In terms of future implications, there has been much evidence suggesting that telehealth use should be continued even after COVID-19 as a support approach for patients and health systems and professionals. Telehealth has shown considerable potential to improve access to healthcare services and facilitate interaction between health professionals and patients; thus, telehealth should be considered in the continuum of quality of care [6,28,29,35,46,74,75]. As health systems continue to deal with the COVID-19 pandemic while also recovering and rebuilding, there is a lot to be learned through this window of opportunity. This creates an occasion for scholars and health system leaders to perform a deep and broad evaluation of what just happened, why, and how can we overcome and predict such crises. With that, telehealth should be at the center of interest considering its importance and role in this pandemic; as we demonstrate in this paper, telehealth has been a huge support for health systems but also has gained a lot of traction due to the COVID-19 pandemic. However, research is still encouraged and always welcome to better understand and examine the opportunities, to document the experiences, successes, and failures of health systems reforms and policy responses in relation to telehealth use during the pandemic leveraging the variations that were seen globally. In addition, policymakers may need to review and adjust and improve their policies regarding telehealth use not only considering ethical questions, but also social and economic inequity and acceptability [28,29,35,46,74].

As a final perspective, we believe that the focus on telemedicine needs to be shifted to developing and under-developing countries [6,74,75,76,77]. In fact, “Equity” as experts see it is vital to incorporate since COVID-19 is more likely to impact on marginalized populations. There is considerable disparity between OECD countries vs. the developing and under-developing nations in terms of telehealth use and progress [73,74,75,76,77]. In several countries, the prompt implementation of telemedicine tools is still currently laborious and even impossible because of a lack of financial and material resources and also knowledge and information [74,75,76]. Therefore, joint initiatives, such as consortiums, and the exchange of competencies and sharing of knowledge in this field between countries are highly recommended [73,74,75,76,77]. In addition, investing in 5G and promoting modern communication technology infrastructures in developing countries will greatly facilitate the implementation of telehealth in developing countries. This might promote a telehealth culture and support governments and populations of those vulnerable countries in implementing concrete solutions, technologies, and policy directives to have these systems available and, of course, decrease this disparity.

## 2. Conclusions

Telehealth is a very interesting approach and can be effective and affordable for health systems aiming to facilitate access to care, maintain the quality and safety of care, and engage patients, health professionals, and users of health services. However, our literature review considers that telehealth faces challenges, such as the issue of lack of human contact in care, confidentiality, and data security, accessibility, training in the use of platforms and telehealth technology, additionally, the disparity that exists in the use of telehealth between countries around the world. However, the article suggests that telehealth is a very promising and reliable approach to help maintain and improve the proper functioning of health services, including in times of global health crises such as the COVID-19 pandemic. Despite the many challenges it faces, telehealth presents an enormous potential for strengthening and improving health services. Additionally, healthcare systems have expressed their great will to maintain and promote the use of telehealth even after the era of the pandemic. Further studies are encouraged to build a solid and broad understanding of telehealth challenges with its short-term and long-term clinical, organizational, socio-economic, and ethical impacts and propose solutions for continuous improvement.

## Data Availability

The data presented in this study are available on request from the corresponding author.

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
