# Peer review of "Telehealth and COVID-19 Pandemic: An Overview of the Telehealth Use, Advantages, Challenges, and Opportunities during COVID-19 Pandemic"

_healthcare, 2022, doi:10.3390/healthcare10112293_

Round 1
Reviewer 1 Report
Thank you for allowing me to review this viewpoint. However, I have shared my opinions.
The abstract is well written. However, it could be better to use “viewpoint” instead of “research paper” as it is not a full-length research paper. Besides, you have written “Re-search Gate” but it will be “ResearchGate”.
In the “context” part, it will be better, if you mention how many papers you have followed or collected from which sources to complete this viewpoint.
The rest of the parts are pretty okay.
Thanks.
Author Response
Dear reviewer Thank you so much for your valuable comments and suggestions your paper can only be better and stronger with your precious review.
Below are our responses in red color.
Reviewer 01:
Comments and Suggestions for Authors
Thank you for allowing me to review this viewpoint. However, I have shared my opinions.
The abstract is well written. However, it could be better to use “viewpoint” instead of “research paper” as it is not a full-length research paper. Besides, you have written “Re-search Gate” but it will be “ResearchGate”.
Yes, This has been corrected now we use the word “paper” instead of “research paper” since a viewpoint/commentary is also considered paper and manuscript.
In the “context” part, it will be better, if you mention how many papers you have followed or collected from which sources to complete this viewpoint.
Yes, As we state in the context section this viewpoint was supported by consultation of many resources but we did not do a systematic or literature review. So, we are unable to precisely give the number of articles as our goal was only to support our commentary and not used to present a literature review. However, we added a section in the context to specify the search words used and also mentioned that this is not a literature or systematic review but only a viewpoint paper.
Thank you so much. We hope our answer will be helpful.
Regards,

Reviewer 2 Report
Thank you for writing this paper. Despite its important topic, I have some concerns as follows:
1. The contributions of this paper are questionable. The authors didn't clearly discuss the states of the art in the current literature, as well as the research gaps that the authors want to address in this paper. For example, I found some literature related to this paper:
- Garfan, S., Alamoodi, A. H., Zaidan, B. B., Al-Zobbi, M., Hamid, R. A., Alwan, J. K., ... & Momani, F. (2021). Telehealth utilization during the Covid-19 pandemic: A systematic review. Computers in biology and medicine, 138, 104878.
- Hincapié, M. A., Gallego, J. C., Gempeler, A., Piñeros, J. A., Nasner, D., & Escobar, M. F. (2020). Implementation and usefulness of telemedicine during the COVID-19 pandemic: a scoping review. Journal of primary care & community health, 11, 2150132720980612.
- etc.
It is imperative to state the position of this research compares to others to show its significance.
2. I also feel that the organization of this paper should be changed. In the current form, it only contain two sections: Context and Conclusion. I suggest the authors to use at least a IMRAD standard (Introduction, Method, Results, and Discussion).
3. The research method is missing. How did you perform the literature review? Did you use a particular method for literature review such as PRISMA? This paper didn't use a particular literature review method. Also, there some questions related to literature review procedure:
- Please check the source of literature review. Why did you only include 4 databases as the sources for retrieval? Have you checked that those databases include reputable journals in this field?
- What are the keywords used to retrieve papers from the databases?
- What are the inclusion and exclusion criteria?
- How did you ensure the quality of the papers being reviewed?
- How did you analyze the papers?
4. For me, the results are too descriptive. I suggest the authors perform thematic analysis to identify themes emerged in the literature. Summary of themes in form of table will help readers in understanding the key findings in this research.
5. I also ask the authors to discuss the implications of this study, both for practice and theory.
Author Response
Dear reviewer Thank you so much for your valuable comments and suggestions your paper can only be better and stronger with your precious review.
Below is our answer to your comments.
1) The contributions of this paper are questionable. The authors didn't clearly discuss the state of the art in the current literature, as well as the research gaps that the authors want to address in this paper. For example, I found some literature related to this paper:
As we state in the context section this is not a research paper this is a paper in a form of viewpoint/commentary. We developed a broad reflection on the topic and the paper is not guided by any form of structured literature review.
We did not discuss the state of the art in the current literature, as well as the research gaps because to do so a systematic and structured literature review needs to be completed and this is not the goal of this paper. Again, this paper is not in the category of a literature review or systematic review paper. This is a simple viewpoint in which we develop a broad reflection of the topic. Still, this viewpoint was supported by consultation of many resources. There was no systematic or literature review.
2) It is imperative to state the position of this research compared to others to show its significance.
Again, this paper is not a research paper and it is not in the category of a literature review or systematic review paper. This is a simple viewpoint/commentary in which we develop a broad reflection of the topic. Still, this viewpoint was supported by consultation of many resources. There was no systematic or literature review. With that being said we can not compare our paper with other research papers on the topic because simply our paper is not a research paper it is a viewpoint paper and we did not submit this paper to publish in the research category but in the viewpoint and perspective category.
3) I also feel that the organization of this paper should be changed. In the current form, it only contain two sections: Context and Conclusion. I suggest the authors to use at least a IMRAD standard (Introduction, Method, Results, and Discussion).
Dear reviewer, yes definitely the structure you suggest is the standard structure for a research paper. But, our paper is not a research paper and this is a viewpoint paper where ideas and concepts are freely discussed by the authors far from the standard structure of research papers. please kindly see the viewpoints and perspectives papers published and you will find out that the structure of this paper is definitely appropriate for a viewpoint paper.
4) The research method is missing. How did you perform the literature review? Did you use a particular method for the literature review such as PRISMA? This paper didn't use a particular literature review method. Also, there are some questions related to the literature review procedure:
This paper is not in the category of a literature review or systematic review paper. This is a simple viewpoint in which we develop a broad reflection of the topic. Still, this viewpoint was supported by consultation of many resources. There was no systematic or literature review. However, we added a section in the context to specify the search words used and also mentioned that this is not a literature or systematic review but only a viewpoint paper.
5) For me, the results are too descriptive. I suggest the authors perform thematic analysis to identify themes emerged in the literature. Summary of themes in form of table will help readers in understanding the key findings in this research.
Actually, there was no results section at all because again this paper is a general discussion and a viewpoint and a reflection on the topic of telehealth based on our experience and previous research projects we completed and those we referred to in this paper. But at no point we wanted this paper to be considered a research paper and that is why there are no method or result sections.
6) I also ask the authors to discuss the implications of this study, both for practice and theory.
This is not a study or research paper but a viewpoint and reflection through which we discuss general aspects of the telehealth domain and certain implications. We developed certain practical implications in the paper such as advantages, limits, and challenges but the theoretical implications are beyond the goal of this paper since the theoretical framework needs to be founded on the structured study and this paper is not a paper study but only a viewpoint paper.
Thank you so much for your valuable time and consideration.
Regards,

Reviewer 3 Report
Telehealth and COVID 19 pandemic: An overview of the tele- health use, advantages, challenges, and opportunities during COVID 19 pandemic
The abstract is clear and concise.
Giving numerical values about the situation of COVID-19 in the world in the context section may attract the attention of the reader while reading the article.
Page 2-line 52-64-67-72-80-The reference software should be revised. The reference should be rearranged according to the journal rules.
Page 4- line 170-Author comments should also be included.
Author Response
Dear reviewer thank you so much for your valuable time, suggestions, and comments. Your paper can only be better and stronger with your review.
1) Giving numerical values about the situation of COVID-19 in the world in the context section may attract the attention of the reader while reading the article.
We definitely agree with you on that. We did our best to put some numbers and quantitative estimations on certain aspects of telehealth use, especially during the COVID-19 pandemic, However, since this paper was a viewpoint paper with a general reflection on different perspectives of the topic and not a research or study paper, we were unable to put these estimations since it was hard on us to verify because we did not conduct a structured literature review or systematic review that can allow us to go deeper and check the quality and veracity of those estimations.
2) Page 2-line 52-64-67-72-80-The reference software should be revised. The reference should be rearranged according to the journal rules.
Yes, we tried to correct this. we hope this is as you expected.
4) Page 4- line 170-Author comments should also be included.
Yes this has been included.
Thank you so much for your help and support.

Reviewer 4 Report
An interesting paper, reviewing what has already published many times since the pandemic. I would like to see if there is a difference between telehealth and telemedicine. Other authors, as you highlight use the term telehelath to reference prophylaxis and preventative medicine. However telemedicine discusses treatments in response to symptoms.
line 287- who is Telecare- how does this differ?
Line 217- "Always in china"- please explain
Line 242- Telehealth (is this a spelling mistake with a capital T)
Line 286- the discussion could be improved by looking at what devices are available to be used. Mobile phones are far common in lower socio-economic classes than tablets or laptops- should new platforms be focussed on these for point of delivery?
Line 300- what is affordable, to the patient, doctor or health service? Does this include costs of service provider, tariff and devices?
Line 305- to make this review stand out, consider adding something about the what communication levels and devices are available and compare these between examples for countries. For example high income countries use 4G/5G networks with latest phones. From my experience these are not available in lower income countries, where 3G is still common and 10 year old phones are the norm. How does telehealth manage the disparity in communication levels.
Author Response
Dear reviewer thank you so much for your valuable evaluation and suggestion. Our paper can only be better and stronger with your review.
below are our answer in red on your comments:
An interesting paper, reviewing what has already published many times since the pandemic. I would like to see if there is a difference between telehealth and telemedicine. Other authors, as you highlight use the term telehelath to reference prophylaxis and preventative medicine. However telemedicine discusses treatments in response to symptoms.
line 287- who is Telecare- how does this differ?
Indeed Telehealth is defined as the combined use of the internet and information technology for clinical and organizational purposes however telemedicine is the use of technology for clinical purposes only and this means for therapy and exchange of clinical information or clinical intervention and no organization or management application. As an illustration, a health mobile app on a digital mobile device on which patients can meet up virtually with physicians is an example of telehalth. another example of a health mobile app on a digital mobile device on which patients can meet up virtually with physicians and the device transmit the vital signs of the patients to the physician is an example of telemedicine. The idea is that Telemedicine is a compartment of telehealth. and yes telemedicine can be used in the treatment and management of diseases, especially chronic diseases as we mentioned in our paper.
line 287- who is Telecare- how does this differ?
indeed telecare or telehealth is very much the same but depends on the context of use. for example in Canada, we use telecare very much in the USA we use the term telehealth very more than telecare. Also, telecare can be used more in mental health and social services, and telehealth is used in all types of healthcare services. To simplify this Telehealth or telecare both use smart technology for clinical and organizational purposes in the healthcare system.
Line 217- "Always in china"- please explain
This simply means we are still in the context of china or we are still talking about this concept or element that took place in china.
Line 286- the discussion could be improved by looking at what devices are available to be used. Mobile phones are far more common in lower socio-economic classes than tablets or laptops- should new platforms be focussed on these for point of delivery?
Yes, definitely this is a good point! The problem is not only the digital device or the smart device the problem we pointed out there is the telehealth system. Indeed even if you have a smartphone or smart tablet but your health system does not offer telehealth platforms, then you have nothing. so when talking about access is not only the device that the patient should have but most importantly the telehealth platform that supports your smart device. Because health platforms are very expensive systems to implement, maintain and always improve and it is not only a question of smartphones or tablets. Besides, we also need very good internet and secure trustable data system and these should also be considered.
Line 300- what is affordable, to the patient, doctor, or health service? Does this include costs of service provider, tariff, and devices?
Affordable in terms of long-term investment. this means the cost of implementing a telehealth platform can be expensive but in the long term, the health system can save money with all the benefits of better health outcomes for the population, better management of chronic diseases, etc. So here affordable does not mean cheap, it means you invest in telehealth platforms and you save money on all chronic disease expenses that are exacerbated by low health access. and since telehealth improves health access then telehealth will be an affordable and good investment in the long term.
Line 305- to make this review stand out, consider adding something about the what communication levels and devices are available and compare these between examples for countries. For example high income countries use 4G/5G networks with latest phones. From my experience these are not available in lower income countries, where 3G is still common and 10 year old phones are the norm. How does telehealth manage the disparity in communication levels.
yes, this has been added in lines 333 to 337. thank you so much for your precious insight.
Thank you so much for you suggestion and review.
Regards,

Round 2
Reviewer 2 Report
I have no more comments